

# Risk-sensitive foraging does not explain condition-dependent choices in settling reef fish larvae

Emma E. Bogdan[1], Andrea L. Dingeldein[1], Deirdre Bertrand[1] and Will White[2]

[1] Department of Biology and Marine Biology, University of North Carolina at Wilmington, Wilmington, NC, United States of America
[2] Department of Fisheries and Wildlife, Coastal Oregon Marine Experiment Station, Oregon State University, Newport, OR, USA

## ABSTRACT

The transition from the planktonic larval to the benthic adult stage in reef fishes is perilous, and involves decisions about habitat selection and group membership. These decisions are consequential because they are essentially permanent (many fish rarely leave their initial settlement habitat, at least for the first several days or weeks). In one common Caribbean reef fish, the bluehead wrasse (*Thalassoma bifasciatum*), settling larvae either join groups or remain solitary. Grouped fish have lower mortality rates but slightly slower growth rates, and fish that are smaller at the time of settlement are less likely to join groups. We hypothesized that the decision of smaller (i.e., lower condition) fish to remain solitary could be explained by risk-sensitive foraging: with less competition, solitary fish may have higher variance in foraging success, so that there is a chance of a high payoff (outweighing the increased mortality risk) despite the lack of a large difference in the average outcome. We tested this by comparing the mean, standard deviation, and maximum number of (a) prey items in stomach contents and (b) post-settlement growth rates (from otolith measurements) of solitary and grouped fish during two settlement pulses on St. Croix, US Virgin Islands. However, we did not find evidence to support our hypothesis, nor any evidence to support the earlier finding that fish in groups have lower average growth rates. Thus we must consider alternative explanations for the tendency of smaller fish to remain solitary, such as the likely costs of searching for and joining groups at the time of settlement. This study reinforces the value of larval and juvenile fish as a testbed for behavioral decisionmaking, because their recent growth history is recorded in their otoliths.

## INTRODUCTION

When making choices that affect fitness in a stochastic environment, animals often account for both the *average* fitness payoff for different alternatives as well as the relative *variance* associated with those payoffs (*Caraco, Martindale & Pulliam, 1980*; *Barkan, 1990*; *Kacelnik & Bateson, 1996*; *Houston & McNamara, 1999*; *Kacelnik & Mouden, 2013*). For example, in the classic original experiment, *Caraco, Martindale & Pulliam (1980)* showed that well-fed

Corresponding author
Will White, jwilsonwhite@gmail.com, will.white@oregonstate.edu

yellow-eye juncos (*Junco phaeonotus*) were risk-averse in their food preferences, preferring feeding stations with a lower variance in the amount of food delivered, regardless of the average amount. However, juncos on a poorer diet were risk-prone, choosing higher-variance feeding stations. This behavior can be explained by the 'budget rule': if an animal's energy budget is sufficient to meet immediate needs (e.g., overnight survival), it will be risk-averse, minimizing the chance of low or zero payoffs. If, however, the energy budget is lacking, the animal will choose the higher-variance option (even if the mean payoff is insufficient for its needs), improving its chance at a life-saving high payoff (*Stephens, 1981*; *Smallwood, 1996*; *Houston & McNamara, 1999*). After those initial experiments in bird model systems, the concept of risk-sensitive foraging has been applied to a wide range of taxa (*Kacelnik & Bateson, 1996*).

This simple version of the budget rule has been criticized for failing to adequately explain experimental data on foraging animals (*Bateson, 2002*; *Kacelnik & Mouden, 2013*), although more sophisticated versions of the rule produce better fits to data (*Lim, Wittek & Parkinson, 2015*). Nonetheless, there is a general expectation that animal behaviors reflect differences in the variance of payoffs from different choices. For example, some spiders switch between sit-and-wait and mobile hunting strategies depending on the variance in prey encounter rates (*Caraco & Gillespie, 1986*; *Gillespie & Caraco, 1987*; but see *Smallwood, 1993* for an alternative explanation). In common eiders, *Somateria mollissima*, birds in poor energetic condition joined smaller flocks and foraged in habitats with less-preferred prey but a more variable energetic return, apparently minimizing competition and gaining the possibility of a bigger payoff in prey collection (*Guillemette, Ydenberg & Himmelman, 1992*).

Many benthic marine organisms face a period of crucial and irreversible decision-making when they make the transition from a highly dispersive planktonic larval stage to a less mobile, benthic adult stage, often with home ranges on the scale of meters or even centimeters (*Doyle, 1975*; *Stamps, Krishnan & Reid, 2005*). The adult habitat selected by the settling larva will have long-term fitness consequences, leading to strong selective pressure for the evolution of adaptive settlement behaviors. For example, larval barnacles use chemical cues from intertidal organisms that share a similar range of environmental tolerances, allowing them to select appropriate locations for settlement in the intertidal (*Raimondi, 1988*). Larval coral reef fish also respond to chemical cues, improving their chance of settling in higher-quality habitats (*Dixson, 2011*; *Dixson, Abrego & Hay, 2014*) and some species also avoid locations that are already occupied by competitors or older conspecifics in order to avoid competition (*Stier & Osenberg, 2010*). All of these examples describe scenarios in which larvae respond to differences in the mean payoff between settlement sites. In this paper, we investigated whether larvae also respond to the variance in fitness payoffs when making settlement decisions.

In addition to the decisions about settlement habitat that other coral reef fishes make, settling larvae of the bluehead wrasse, *Thalassoma bifasciatum*, also face a choice about social group membership. Bluehead wrasse are one of the most common fish on Caribbean reefs, and adults are highly mobile, swimming rapidly around the reef in loose aggregations. This species is also well-known as a model system for investigations of the evolution and
behavioral ecology of mating systems and protogynous sex change (*Warner, 2001*). Here we focus instead on the behavioral ecology of the early juvenile stages. At the time of settlement from the plankton, larval bluehead wrasse bury themselves in the reef sediment for approximately 3–5 days while they metamorphose (*Victor, 1982*). When they emerge, juvenile wrasse are highly site-attached for the first week of their life on the reef, staying within tens of cm from a shelter crevice while cautiously feeding on zooplankton in the water column. At this time, juvenile bluehead wrasse are either solitary or form small groups of up to twenty. Fish that are larger at the time of settlement are more likely to be found in groups, and per-capita mortality declines with increasing group size (*White & Warner, 2007a*; *Dingeldein & White, 2016*). *White & Warner (2007b)* showed that fish in groups spend considerably more time foraging than solitary fish, but have somewhat slower post-settlement growth rates, likely due to competition. Fish vary in size and age within groups (although all are less than approximately 7 post-emergence days old), and aggressive 'chasing' encounters are common among groupmates (*White & Warner, 2007b*). Bluehead wrasse begin to become less site-attached and leave these groups after approximately one week, changing behaviors to associate with mobile shoals of older, larger conspecifics. However, that first week on the reef is a window of very high mortality, and a time when behavioral decisions alter mortality risk considerably.

These observations of behavior and growth suggest that smaller fish trade the safety of group membership for the opportunity for a faster growth rate (*White & Warner, 2007a*; *White & Warner, 2007b*; *Dingeldein & White, 2016*). Faster growth is a metric of (eventual) fitness in small immature fish. This is because many coral reef predators are gape-limited (they can only consume things smaller than their mouth opening) so as small fish grow, fewer predators are able to consume them. Thus faster-growing fish spend less time in vulnerable size classes, conferring greater survival and a better chance of reaching reproductive age (*Miller et al., 1988*; *Houde, 1989*). However, the negative relationship between growth rate and group size reported by *White & Warner (2007b)* was small, and perhaps not biologically significant (though statistically significant, the $r^2$ was only 0.09). Therefore, we investigated whether the group-joining decision of juvenile bluehead wrasse was risk-sensitive, and a response to the variance in fitness outcomes rather than (or perhaps in addition to) the mean.

We hypothesized that small fish may be more likely to remain solitary because of the potential for higher prey capture rates and higher growth rates. To test this hypothesis, we re-analyzed the dataset collected by *Dingeldein & White (2016)*, who found an effect of size-at-settlement on the decision to join groups, but did not examine the post-settlement growth rates of the fish they collected. We analyzed the post-settlement growth rates (estimated from otolith growth rings) to test for differences between solitary and grouped fish in the (a) mean and (b) variance of both gut fullness and growth rates. We anticipated that while the means would not differ (or differ only slightly), solitary fish would exhibit higher variances in growth, indicating that remaining solitary is a risk-prone strategy for small juvenile wrasse.

## MATERIALS & METHODS

The samples used in this study were collected by *Dingeldein & White (2016)*, and additional details of collection are provided there. Recently settled juvenile bluehead wrasse were collected using hand nets and clove oil anesthetic from three sites on the northwest shore of St. Croix, USVI (Fig. S1). Bluehead wrasse settle to the reef in approximately week-long pulses following a new moon (*Caselle & Warner, 1996*); collections for this study occurred during settlement pulses in July and August of 2012. *Dingeldein & White (2016)* described collecting two sets of fish: zero-day collections, in which larvae settling to a transect were collected on their first day after emergence onto the reef, and additional collections in which entire groups and solitary fish were selected haphazardly for collection after they had been on the reef for 1–4 days (age could be ensured because the transects were cleared of all fish on day 0, and tagging has shown that fish do not move between shelter crevices after emergence; *White & Warner, 2007a*). We used the latter set of collections to examine patterns of post-settlement growth. Fish were preserved immediately after each dive in 75% ethanol.

All samples were collected following the current laws of the United States Virgin Islands (USVI); fieldwork was performed in accordance with the USVI Department of Planning and Natural Resources (Permit No. STX-041012) and with approval of the University of North Carolina Wilmington's Institutional Animal Care and Use Committee (Protocol A1011-009), in compliance with the US National Research Council's Guide for the Care and Use of Laboratory Animals.

### Planktonic resource quantification

To quantify the availability of the bluehead wrasses' planktonic prey, we conducted plankton tows on SCUBA at each site, swimming approximately 0.5 m over the reef, perpendicular to the transects on which fish were collected. The width of the transect area (∼30 m) was sampled twice by beginning at the first transect, swimming out to the last, and returning to the beginning. Plankton tows were conducted on the same days that fish were collected (except for the first of four days of sampling in both July and August at the Butler Bay site). Plankton samples were filtered through a 150 μm sieve, fixed in 10% formalin, and preserved in 75% ethanol. A 1 mm$^2$ gridded Sedgewick-Rafter cell was used to count the number of cyclopoid, harpactacoid, and calanoid copepods (and several other taxonomic groups) present in 1 mL of each sample. These counts were scaled up to obtain abundance estimates for the entire sample. A flowmeter was attached to the front of the plankton net to obtain volumetric measurements of the amount of water that was sampled on each tow. This provided an estimate of the amount of available prey/m$^3$ present in the water column at each given site and day.

### Otolith analysis

After preservation, sagittal otoliths were extracted from each fish and placed in microscope immersion oil for at least thirty days prior to improve clarity. We photographed whole otoliths at 400× under polarized light using Leica Acquire 1.0 software (Leica Microsystems, Buffalo Grove, IL, USA). We counted and measured daily otolith increment widths using

ImageJ software (National Institutes of Health, Bethesda, MD, USA), starting at the first visible ring and counting along the longest axis (post-rostrum). In bluehead wrasse, the timing of both initial larval settlement and subsequent emergence onto the reef is clearly demarked on the otolith by a wide metamorphic band (*Victor, 1982*). Therefore we were able to measure both post-settlement age (number of bands after the metamorphic band) and post-settlement growth rate (the mean width of post-settlement increments). Each otolith was read by the same two people and the results were compared; otoliths were measured again if the post-settlement age did not agree, and discarded if the readers could not reach an agreement. Data were also discarded if the metamorphic band width (MBW) measurements differed by >10%.

### Diet analysis

Stomachs of each preserved fish were dissected under 10x magnification to estimate diet composition and stomach fullness at the time of collection. Juvenile bluehead wrasse feed continuously from approximately 30 min. after dawn until approximately 30 min. before dusk (JW White, pers. obs., 2002), and all fish in this study were collected at least two hours after dawn and two hours before dusk to avoid crepuscular periods when the fish may have changed behavior to avoid predation risk. Diet items were classified to the lowest taxonomic level possible (usually order) and counted. Most diet items were clearly identifiable planktonic or benthic crustaceans (copepods, isopods, amphipods) and, following *White & Warner (2007b)*, fullness was estimated as the total number of items in the stomach.

### Statistical analysis

To examine differences in mean and variation in diet and post-settlement growth as a function of group size, we treated each group as an individual replicate and calculated the mean, standard deviation, and maximum number of diet items and post-settlement growth rates observed in each group. We examined the maximum because the rationale of risk-sensitive foraging is that a risky strategy affords a potentially greater fitness payoff despite a similar or lower mean fitness payoff. Solitary fish on a given reef and day were also considered to be a replicate 'group' (with a group size = 1) for the purposes of calculating these statistics.

We used linear models to test for an effect of group size on mean, standard deviation, and maximum number of diet items and post-settlement growth. We performed two separate tests for each of the response variables. First, we tested for a continuous effect of group size (e.g., growth rate declines with each additional group member), so the main effect was 1/(group size). Second, we tested for a simple binary difference between solitary fish and grouped fish by pooling all group sizes together. In each model we also included fixed effects of site and month to account for possible spatiotemporal covariation in growth rates, as well as a fixed effect of planktonic copepod density (copepods were the dominant prey item in fish stomachs; see 'Results'). We removed those effects from reported model results in a backwards stepwise fashion if their effects were clearly not statistically meaningful ($p > 0.2$). For diet analyses, the planktonic copepod covariate was simply the density of

copepods on the day the fish was collected (only fish collected on days when plankton tows were made were included in analyses with that covariate). For growth analyses, the effect of resource availability would be integrated over the post-settlement life of the fish. Therefore, for each fish, we calculated the average copepod density during the prior days the fish had been on the reef (based on the estimate of post-settlement age). We then averaged those hindcast copepod density estimates for all of the fish in each group. For diet analyses, the distributions of the mean and maximum number of diet items were asymmetrical, with long tails, so we applied a $\log(x+1)$ transformation to the data prior to analysis. All other response variables met the distributional assumptions of linear models.

*Dingeldein & White (2016)* had reported that fish that were larger at the time of emergence onto the reef (estimated from the otolith axis radius from the core to the outer edge of the metamorphic band) were more likely to join groups. Consequently we were concerned that size at emergence could subsequently confound detection of group effects on growth, if larger fish also tended to grow faster. We tested for this relationship and found that although it was statistically significant, due primarily to the very high sample size ($p = 0.04$, $df = 228$; Fig. S2), it had essentially no explanatory power ($R^2 = 0.01$). We therefore decided that there was little risk of confounding effects from this factor.

When the group size effect was found to be not statistically significant in the reduced linear models, we evaluated the statistical power of the test. We did this by estimating the power of the $t$ statistic associated with the group size regression coefficient (in a two-sided test context). For these analyses, we fixed the variance of the statistic at the level observed in the test, and then calculated power for a range of sample sizes and effect sizes using the pwr package in R (*Champley, 2018*).

All analyses were conducted using R 3.5.1 (*R Development Core Team, 2018*); statistical models were constructed using the function lm in the basic R installation. Data and code are available at github.com/jwilsonwhite/bluehead_risk_sensitivity. Graphics were produced using ggplot2 (*Wickham, 2016*).

## RESULTS

After sample processing and quality control, we were able to analyze diet and otolith data from 230 fish that were ≥ 1 day post-settlement age (allowing a calculation of post-settlement growth rate). These fish comprised 97 individual groups of ≥ 2 fish and 25 'groups' of solitary fish (i.e., all of the solitary fish collected from a site on a given day). Settlers ranged in age from 1 to 7 days post-settlement, though 95% of the individuals were ≤ 4 days post-settlement age. The distribution of post-settlement ages in the collection did not differ between grouped and solitary individuals (Fig. S3). When calculating the mean and maximum number of diet items and growth rates, we used only data from 'groups' for which ≥ 2 individuals were collected and successfully processed (total $n = 49$; this comparison included the 'groups' of solitary fish); when calculating the standard deviation we were more conservative and only used data from groups for which ≥ 3 individuals were available (total $n = 28$, again including 'groups' of solitary fish). The latter rule necessarily excluded all pairs of fish (group size = 2) but ensured that estimates of variance in each

group had at least $n = 3$ (it did not include 'groups' of solitary fish comprised of $\geq 3$ fish collected at the same site and date). The maximum number of individuals analyzed from any one group was 6.

## Planktonic resource availability

The density (number/m$^3$) of copepods sampled in plankton tows varied over two orders of magnitude between July and August across all sites, and was also variable (though less so) from day-to-day at each site (Fig. S4).

## Diet

The overall diet composition of fish examined was primarily harpacticoid, cyclopoid, and calanoid copepods (65%, Fig. S5), with the remainder consisting of amphipods, isopods, ostracods, foraminifera, bivalves, and gastropods. Because copepods were both the most frequently occurring and the largest, most energy-rich prey items (cf. *White & Warner, 2007b*), we focused our analyses on the numbers of copepods only.

There was no effect of group size (modeled as 1/[group size]) on the mean, standard deviation, or maximum number of copepods in fish stomachs (Fig. 1, Table S1). For both the mean and maximum, the effect of month was not significant but did not meet the threshold for stepwise removal ($0.05 < p < 0.2$), so that effect is depicted in Fig. 1 as a trend towards more diet items in stomachs during the second month of sampling. For the standard deviation of stomach items, there was a significant positive effect of planktonic copepod abundance, and fish at the Northstar site had significantly fewer prey items in their stomachs. The site effect is depicted in Fig. 1B, which displays the residual standard deviation with the effect of planktonic copepods removed. We obtained nearly identical results when group size was modeled as a binary factor (solitary vs. grouped; Table S1, Fig. S6).

In general, the effects of group size were in the direction we hypothesized (e.g., greater standard deviation in solitary fish) but observed effect sizes were low (e.g., 7% lower mean and 4% lower standard deviation in a group of two relative to solitary fish), variation was very high (Fig. 1), and the patterns were clearly not statistically meaningful ($p > 0.5$ for all group size effects). We assessed our power to detect any effect, given the variation in our response variables (Fig. S7). For the mean and the maximum, power would not be $>0.8$ for the observed effect size even if sample size were quadrupled to over 100 groups of fish. For standard deviation, increasing power to 0.8 would have required nearly quadrupling sample size to 50 groups of fish (recall that we had a smaller sample size for that analysis).

## Growth rate

There was no effect of group size (modeled as 1/[group size]) on the mean, standard deviation, or maximum post-settlement growth rate (Fig. 2, Table S2). There were faster mean and maximum growth rates and smaller standard deviations in growth rate in the first month of sampling (despite lower planktonic resource abundances), and those effects are also depicted in Fig. 2. For standard deviation in growth, the effects of site and planktonic copepod abundance were not significant but did not meet the threshold for removal from the model ($0.05 < p < 0.2$), so the site effect is shown in Fig. 2B and the response variable

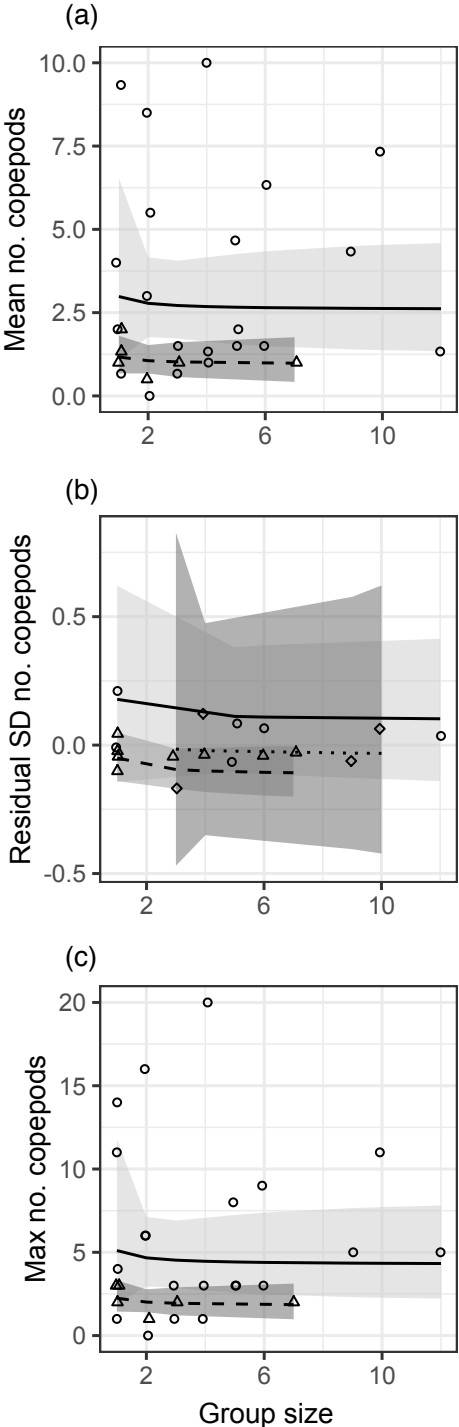

**Figure 1** **Relationships between different metrics of the number of copepods in guts of juvenile blue-head wrasse and group size (solitary fish have a group size of 1).** Each data point represents an individual group of fish or the sample of solitary fish on a particular reef and day. Each panel shows a different diet statistic: (A) mean number of copepods in guts within a group; (B) standard deviation of number of copepods within a group; (C) maximum number of copepods within a group. 

**Figure 1 (…continued)**
Lines indicate linear model fits (with group effect modeled as 1/[group size]) and shading indicates 95% confidence region around model fits. In (A, C), the first month (July 2012) is shown as triangle points, dashed curve and darker shading; circles, solid curve and lighter shading denotes the second month (August 2012). In (B), data are displayed as residuals with the effect of planktonic copepod abundance removed, and displayed according to site: Cane Bay (circles, solid curve, light shading), Northstar (triangles, dashed curve, medium shading), and Butler Bay (diamonds, dotted curve, dark shading).

is shown as residuals with the effect of planktonic copepod abundance removed (as in Fig. 1B). We obtained nearly identical results when group size was modeled as a binary factor (solitary vs. grouped; Table S2, Fig. S8).

The effects of group size on growth varied, with slightly positive (but not significant) effects of group size on the mean and maximum growth rate (contrary to our hypothesis), but a slightly negative effect (also not significant) of group size on the standard deviation in growth (as we hypothesized). However, the observed effect sizes were very low (e.g., 9% lower standard deviation in a group of two relative to solitary fish), variation was very high (Fig. 2), and clearly not statistically meaningful ($p > 0.25$ for all group size effects). We assessed our power to detect any effect, given the variation in our response variables (Fig. S9). For the mean and the maximum, power would not be >0.8 for the observed effect size even if sample size were quadrupled to over 100 groups of fish. For standard deviation, power was 0.95 at the observed effect size, variance, and sample size.

## DISCUSSION

The goal of this study was to determine whether group-joining decisions by settling fish larvae could be explained in terms of risk-sensitive behavior. Prior research had shown that fish that were larger at settlement were more likely to join groups (*Dingeldein & White, 2016*), and that larger groups of juvenile bluehead wrasse had higher per capita survival but slower growth rates (*White & Warner, 2007a*; *White & Warner, 2007b*). We extended that earlier work by examining the variation in growth rates within entire groups of fish. Contrary to the trend reported by *White & Warner (2007b)*, we found no effect of group size on mean growth rate. Additionally, we did not find support for our hypothesis that solitary fish have higher variation in feeding rate and growth rate than do grouped fish. Thus we find no support for risk-sensitive foraging behavior as an explanation for the observation that smaller fish are more likely to remain solitary (*Dingeldein & White, 2016*).

These results suggest that bluehead wrasse that join groups at the time of settlement experience lower predation risk, despite spending more time foraging in the water column (*White & Warner, 2007a*; *White & Warner, 2007b*), with no apparent cost in terms of post-settlement growth. This is reinforced by our finding that there was not a relationship between post-settlement growth rate and fish size at settlement. Evidently, the latter trait (which is shaped by the larval origin and dispersal trajectory of the fish; *Hamilton, Regetz & Warner, 2008*) affects the propensity to join groups (and thus mortality risk) but not post-settlement growth. Why did we not find the same pattern of mean post-settlement growth as *White & Warner (2007b)* did, at some of the same study sites? The most likely explanation is that negative relationship reported by *White & Warner (2007b)*

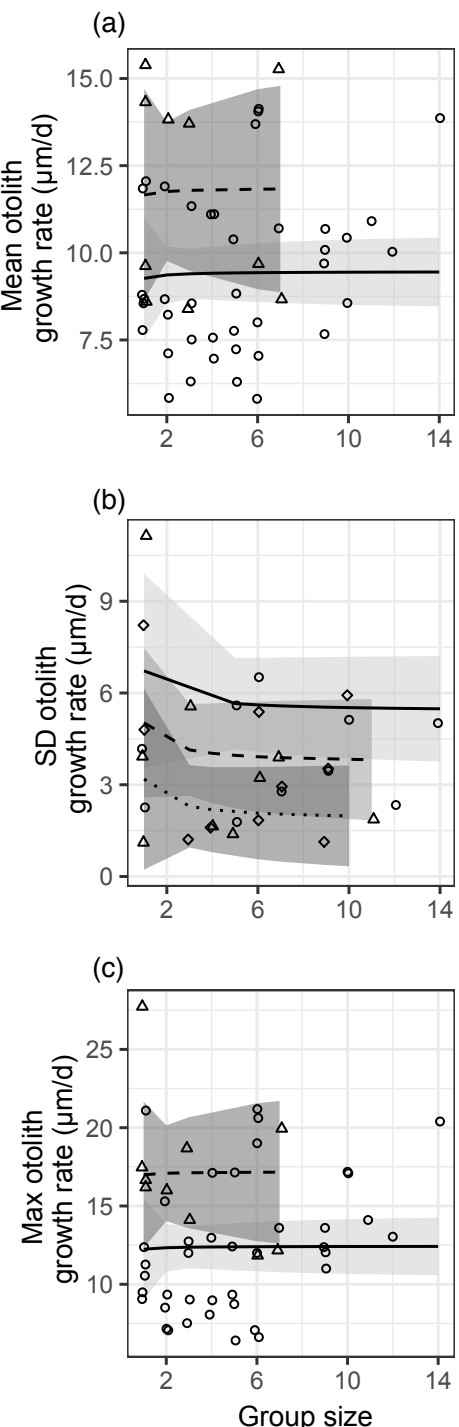

**Figure 2** **Relationships between different metrics of juvenile bluehead wrasse post-settlement growth rates (measured in otoliths) and group size.** Each data point represents an individual group of fish or the sample of solitary fish on a particular reef and day. Each panel shows a different growth rate statistic: (A) mean growth rate within a group; (B) standard deviation of growth rate within a group; (C) maximum growth rate within a group. Lines indicate linear model fits (with group effect modeled as 1/[group size]) and shading indicates 95% confidence region around model fits. (continued on next page...)
**Figure 2 (...continued)**
In (A, C), the first month (July 2012) is shown as triangle points, dashed curve and darker shading; circles, solid curve and lighter shading denotes the second month (August 2012). In (B), data are displayed as residuals with the effect of planktonic copepod abundance removed, and displayed according to site: Cane Bay (circles, solid curve, light shading), Northstar (triangles, dashed curve, medium shading), and Butler Bay (diamonds, dotted curve, dark shading).

was slight, and only detectable when variation in planktonic prey resource availability was included as a covariate. It is possible that there is temporal variation in the shape of the relationship, fluctuating between slightly negative and flat, perhaps reflecting variability in the composition of the prey field or other environmental factors that affect energetics and growth.

One unusual aspect of our results was the opposite effects of the sample month on stomach fullness (a trend towards fewer items in the first month than the second, which matches the pattern of abundance in planktonic copepods over each study reef) and post-settlement growth (faster in the first month than the second). This pattern is counterintuitive, and we cannot offer a simple explanation. At the scale of individual fish, the two measures reflect different time scales: stomach contents reflect gut passage time (likely hours), while post-settlement growth integrates multiple days of resource availability. Fish collections were made on multiple days during each monthly recruitment pulse, and planktonic prey availability differed by nearly an order of magnitude at a single site from day to day, so this effect may simply reflect a few high-prey-abundance days in the second month, but mean conditions that did not favor faster growth in that month. In hindsight it would have been preferable to use a sampling approach that integrated copepod abundance over multiple days, as in *White & Warner (2007b)*, but that was not logistically feasible in this study.

When reporting results that fail to reject the null hypothesis, one must consider the evidence that a Type-II error is being made. This would be a particular concern if marginal increases in either effect size or sample size might have produced substantial increases in power, meaning that repeating the study or increasing sample sizes would yield significant results. Though we acknowledge the potential problems with post-hoc power analysis (e.g., *Underwood, 1999*), in this case our power analyses suggest that in most cases the observed effect sizes were simply very small relative to the variance in response variables, such that even drastic ($4\times$) increases in sample size would not have yielded meaningfully higher power. The exception was for our test of a group size effect on the standard deviation of growth rates, which had power $>0.9$, also supporting our conclusion that we did not commit Type-II error. This is reinforced by examining the data in Figs. 1 and 2: the distributions of data for every metric of both diet and growth rates overlap considerably across group sizes, and differences in central tendency are very small relative to the variability in the response variables. Based on that evidence, we doubt that we would have detected any meaningful statistical results with greater sample size.

Given the lack of evidence for risk-sensitive foraging, we turn to an alternative hypothesis for the tendency of smaller fish (at settlement) to remain solitary. *Stamps (2006)* proposed the 'silver spoon' hypothesis for habitat selection by dispersing juveniles. This hypothesis

has two parts: individuals in better condition can (a) afford to be choosier during habitat selection, searching longer to find better habitat, and (b) better compete for a contested location, or for membership in a group that might attempt to reject them. It is reasonable to see how this could apply to coral reef fish; larvae that have just settled onto the reef (or emerged from the sediment post-metamorphosis, in the case of bluehead wrasse) must find a suitable shelter habitat (and group) quickly, because traversing the reef during a search carries high predation risk. A larger fish would have faster swimming speed and thus be able to search more area without incurring additional predation exposure. We have never observed eviction from groups of juvenile bluehead wrasse, so part (a) of the hypothesis appears to be more relevant than part (b), at least in this species. Of course, testing such hypotheses are challenging because larval settlement behaviors happen at night in unpredictable locations (and when the animals are small and nearly transparent), and they are difficult to study (*Holbrook & Schmitt (1997)* is the only example of which we are aware). However, it may be possible to examine the relative contribution of instantaneous mortality risk during the search and deferred mortality risk in subsequent days after habitat and group selection is complete, using a modeling approach like that of *Stamps, Krishnan & Reid (2005)*. An additional possibility to consider is that the solitary versus grouped behaviors reflect different fish personalities, particularly variation in the relative boldness of individuals (*Biro & Stamps, 2008*).

## CONCLUSIONS

Larval fish are a rich testbed for examining the influences on behavioral decision-making, because they carry in their otoliths a record of their past condition and growth history (*Booth & Beretta, 2004*; *Grorud-Colvert & Sponaugle, 2006*; *Dingeldein & White, 2016*). The details of how larvae make habitat-selection and group-joining decisions continue to be a topic of considerable interest and investigation (e.g., *Stier & Osenberg, 2010*). For bluehead wrasse, we had hypothesized—based on prior studies—that the likely explanation for the highly consequential decision to join a group or not was based on the potential for higher fitness payoffs for solitary fish. However, our data did not provide any support for that hypothesis, and the observed ratios of signal to noise suggest that this conclusion was not due to a lack of statistical power. We hope that future studies may shed more light on the selective factors underlying these behavioral decisions.

## ACKNOWLEDGEMENTS

We thank M Heintz for assistance in the field, and A Orpen, J Jaramillo, C Brady, and L Lukas for assistance in the lab. St. Croix Ultimate Bluewater Adventures provided valuable logistical support. DLB thanks her father, the late RJ Bertrand, and her son, A Bertrand, for their enduring support. L Tuttle and an anonymous reviewer provided helpful suggestions that improved the manuscript.

### Funding

This work was supported by the University of North Carolina Wilmington Department of Biology and Marine Biology. The funders had no role in study design, data collection and analysis, decision to publish, or preparation of the manuscript.

### Grant Disclosures

The following grant information was disclosed by the authors:
University of North Carolina Wilmington Department of Biology and Marine Biology.

### Competing Interests

Will White's partner, Susanne Brander, is an Academic Editor for PeerJ.

### Author Contributions

- Emma E. Bogdan and Deirdre Bertrand performed the experiments, authored or reviewed drafts of the paper, and approved the final draft.
- Andrea L. Dingeldein conceived and designed the experiments, performed the experiments, authored or reviewed drafts of the paper, and approved the final draft.
- Will White conceived and designed the experiments, analyzed the data, prepared figures and/or tables, authored or reviewed drafts of the paper, and approved the final draft.

### Animal Ethics

The following information was supplied relating to ethical approvals (i.e., approving body and any reference numbers):

The University of North Carolina Wilmington's Institutional Animal Care and Use Committee provided full approval for this project, in compliance with the U.S. National Research Council's Guide for the Care and Use of Laboratory Animals (UNCW IACUC Protocol A1011-009).

### Field Study Permissions

The following information was supplied relating to field study approvals (i.e., approving body and any reference numbers):

Fieldwork was performed in accordance with the United States Virgin Islands Department of Planning and Natural Resources (Permit No. STX-041012)

### Data Availability

Data and code are available at GitHub: https://github.com/jwilsonwhite/bluehead_risk_sensitivity.

### Supplemental Information

Supplemental information for this article can be found online at http://dx.doi.org/10.7717/peerj.8333#supplemental-information.

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
