# Peer review of "Risk-sensitive foraging does not explain condition-dependent choices in settling reef fish larvae"

_PeerJ, doi:10.7717/peerj.8333_

## Round 0.1 · original submission · Minor Revisions

Overview
This manuscript tests a hypothesis to explain an earlier finding that smaller juvenile bluehead wrasses are less likely to join groups than larger juveniles. The hypothesis is that this pattern reflects a risk-sensitive foraging decision (solitary fish were expected to have higher variability of feeding rate and growth). Using data on food intake and growth rate obtained from previously collected specimens, the authors found that the hypothesis was not supported; a power analysis indicated that this was unlikely to be the result of an inadequate sample size. The manuscript is well-written and the reviewers agree that it is a sound study.

Editor’s Comments

You should respond to all reviewer comments by either making appropriate changes or providing a clear explanation for why this should not be done. I would like to provide some additional notes regarding some of these suggestions.

Reviewer 1 requests several clarifications of the behavioral/population context of your study, as well as some aspects of the Methods and writing style. I would particularly like to emphasize her request for more information on the context of the study to help readers not familiar with the system. Some of the concerns raised by Reviewer 2 also reflect this lack of information. Because I have studied coral reef fish in the Caribbean and am familiar with the research of Bob Warner and many of those who have carried out research with him, I had failed to realize how much relevant information was missing. A brief description of the behavior of solitary fish and those in groups as well as information about changes as the fish grow would be useful. Perhaps you could try your revised description on some readers unfamiliar with the system to see if you have successfully conveyed a sense of the behavioral context of your study.

The most critical comment from Reviewer 2 is that you failed to include solitary fish in the analysis. I did not have this impression in my reading, so went back to assess the discrepancy between our views. I believe that you did include solitary fish in your analysis. However, I found that the explanation that you included all solitary fish on a given reef and day as a group size of 1 (L189-190) lacked clarity. The statement that you included data only from groups of 2 or greater (L234) added to the confusion. The grouping of solitary fish concept was somewhat clarified in the Results (L230) but needs to be much more explicit in the Methods.

L134. If the collected fish were truly selected at random, briefly indicate the randomization procedure. If not truly random, the appropriate term is ‘haphazardly’.

Twice I looked for references cited in the text only to discover that they had not been included in the references. Please check all references in the text to be sure that they are all included in the Reference section and that no other references are listed there.

I have included a pdf with a number of other suggested minor changes in word use.

·

Basic reporting

Introduction: I would like to know more about these groups of bluehead wrasse. What is the range of ages of fish in groups? Are the groups exclusively new recruits of the same cohort, more or less, or is there within-group variation in age/cohort? Has there been any documentation of aggression within groups, or descriptions of how groups interact with solitary wrasse? Also, do these solitary wrasse remain solitary their whole lives? If not, at what age do they typically join groups, and could this delay in joining a group tell us something about the putative mechanisms underlying their decision(s)?

Methods: Did you use a particular package in R to create your linear models?

Results: Same as my first question for the Introduction, what is the range of ages of fish in groups? Also, while the number of individuals in a group is visible in the Figures, consider reporting this range in the text.

L72: Change “encouter” to “encounter”
L208-210: Remove “Because” to make this a more coherent sentence.
L332: Change “post-poc” to “post-hoc”
L369: Change “study” to “studies”
Figure 2 is missing an x-axis label.

Experimental design

Methods: How do you know that all fish were collected after they had been active and feeding for at least one hour? Why did you choose to model site and month as fixed effects instead of random effects in a mixed effects model?

Validity of the findings

Discussion: I wonder if the “choice” of smaller fish to join or not join groups is an actual choice by the individual. For instance, do they try to join a group but find it too energetically costly to “keep up”? Are they rejected by groups, as you briefly mention in the Stamps hypothesis on Line 347? Is there any evidence of these kinds of behaviors given your close observations of these fish? There has also been a lot of recent attention paid to animal personalities. Thus, I wonder if the fish that remain solitary are more timid while the fish that join groups are more bold. There are many possible behavioral, physiological, and genetic reasons why you would observe the demographic patterns that you do. You can’t investigate or discuss all of them but consider acknowledging more of them.

Additional comments

I found this study very interesting, well-done, and well-written. I appreciate the power analysis, which improves the interpretation of the results within the context of the study, especially because the study focuses on patterns of variation. I also applaud the authors for pursuing publication of these analyses despite there being negative/inconclusive results. It is an important contribution to their field, nonetheless. In my opinion the manuscript requires only minor revisions.

One additional comment about the authors’ descriptions of animal behavior…

L64-66: “the bird will choose the higher-variance option (even if the mean payoff is insufficient for its needs) in order to have a chance at a life-saving high payoff”
L83-87: “For example, larval barnacles use chemical cues from intertidal organisms that share a similar range of environmental tolerances to select appropriate locations for settlement in the intertidal (Raimondi 1988). Larval coral reef fish also respond to chemical cues to identify higher-quality habitats (Dixson 2011, Dixson et al. 2014) and some species also avoid locations that are already occupied by competitors or older conspecifics in order to avoid competition”

…As a general rule of thumb when one describes animal behavior, one should be careful with the use of the word “to” (or “in order to”). “…in order to have a chance at a life-saving high payoff…” ascribes intent to the animal’s actions. We cannot know the reason why an animal behaves in the way it does, but we can describe what it does without assuming reason/intent. The same can be said for “…to select appropriate locations…”, “…to identify higher-quality habitats…”, and “…in order to avoid competition” in the second excerpt. Luckily, you can either remove these statements of intent, or simply re-word them to be less presumptuous. For example, the first excerpt could become, “the bird will choose the higher-variance option (even if the mean payoff is insufficient for its needs), thus improving its chance at a life-saving high payoff”

Reviewer 2 ·

Basic reporting

I found this article to be well written and contained clear and relevant figures.

Experimental design

Overall, the research fits the aims and scope of the journal and has been conducted to a high standard.

1.I'm not convinced that the plankton sampling was conducted in a manner that is applicable to the experimental design. The authors state that they conducted planton towes over a 30m area for each transect. However, the authors state that these fish stay very close to shelter during the first few days. Sampling plankton in the manner done here will not capture if there was any spatial variation in planton supply between groups.

2. From your introduction you state that these fish spend there first 3 or so days in the sediment while they metamorphose. Can please clarrify if day zero collections (line 132) are fish that have settled to the reef that day or if they are fish that have emerged from the sediment that day. Given the experimental design I'm assuming that you mean day zero means emergency from the sedimental.

Validity of the findings

I'm not convinced that the statistical analysis actually address the stated questions. The experiment was designed to test for differences between solitary and grouped in the mean, max. and SD within groups for gut content and growth rates. This would suggest the analysis should directly contrast grouped vs solitary individuals. However, the authors have used group size as the independent variable and importantly exclude solitary individuals from the analysis (line 234). As such I fail to see how you can draw conclusions based about the decisions of individuals to join groups or not. Additionally, in the figures the authors include the data for the solitary fish. I'm not sure why these were included in the figures if they were not included in the analysis.

Additional comments

In this study the authors state that they aimed to examine whether the decision to join groups or remain solitary could be explained by risk sensitive foraging, where smaller fish would remain solitary and endure higher risk in order to potential gain access better foraging resources. To address this, they tested whether solitary fish and fish in groups differed in their mean, variance or maximal growth rates and food consumption. They found there was no effect of group size on growth rate or any difference between solitary and grouped fishes. My main concern is that as I read the results the authors have not actually tested for the effects of solitary vs grouped fish (more details above). Furthermore, I would suggest that the authors should be cautious in suggesting that the experimental design tested group joining decisions of individuals. Such results could also result from selective predation or from group members rejecting individuals from joining groups (see Coker et al. 2013 MEPS). Overall, I think this paper is of interest but needs either re-analysis of the data or clarification of the current analysis. I have added several minor comments below.


line 177: Please a reference to support that these fish feed continuously throughout the day. I would expect that they do feed continuously but that their feeding rate will vary during the day in line with predations risk (see risk allocation hypothesis and Catano et al. 2017, Oecologia; Feeney et al. 2012 coral reefs). It might be worth adding time of day of collection to your analysis.

Line 61-66: I think this section could with some editing. It currently reads as though the budget rule is specific to birds and thus leaves me wondering why you are discussing it in the context of fish. It would be good to talk about how it relates to all animals and add some additional refs.

Line 233: Can you please provide the statistical analysis for the age distribution of solitary vs grouped fish please. A figure might also be useful.

Line 235: Can you please explain why you only analysed 49 groups when you said you measured 97 groups (line 229)?

Discussion: As an alternative explanation to your results I would look into some of the work done of social groups, as studies have shown that fish tend to associate with similar sized fish in schools (e.g. Hore et al. 2004 Behavioural ecology). It seems to me that larger fish are more likely to encounter groups of similar sized fish than small as the groups are likely to have been feeding on the reef for a day or two and hence be larger than most settling fish. I would think you should be able to test for this with your data set by looking at the variance in size with and between groups.

---

## Round 0.2 · accepted · Accept

The manuscript has been appropriately revised and is now ready for publication. I appreciate the clear and careful response to reviewers letter that made it easy to confirm the changes. Note, however, that some supplemental figures lack suitable captions, for example, dot color in S4, circles vs. triangles in S5, description of the lines in S7 and S9. Please add more complete captions to these figures and check that the others are readily understood by a reader.